# Apoptosis recognition receptors regulate skin tissue repair in mice

Olivia Justynski[1], Kate Bridges[2], Will Krause[1], Maria Fernanda Forni[1], Quan M Phan[3], Teresa Sandoval-Schaefer[1], Kristyn Carter[1], Diane E King[4], Henry C Hsia[5], Michael I Gazes[6], Steven D Vyce[6], Ryan R Driskell[3], Kathryn Miller-Jensen[1,2], Valerie Horsley[1,7]*

[1]Dept. of Molecular Cellular and Developmental Biology, Yale University, New Haven, United States; [2]Dept. of Biomedical Engineering, Yale University, New Haven, United States; [3]Washington State University, SMB, Pullman, United States; [4]Sunnycrest Bioinformatics, Flemington, United States; [5]Dept. of Surgery (Plastic), Yale School of Medicine, New Haven, United States; [6]Dept of Podiatric Surgery, Yale New Haven Hospital, New Haven, United States; [7]Dept. of Dermatology, Yale School of Medicine, New Haven, United States

*For correspondence:
valerie.horsley@yale.edu

**Abstract** Apoptosis and clearance of apoptotic cells via efferocytosis are evolutionarily conserved processes that drive tissue repair. However, the mechanisms by which recognition and clearance of apoptotic cells regulate repair are not fully understood. Here, we use single-cell RNA sequencing to provide a map of the cellular dynamics during early inflammation in mouse skin wounds. We find that apoptotic pathways and efferocytosis receptors are elevated in fibroblasts and immune cells, including resident Lyve1+ macrophages, during inflammation. Interestingly, human diabetic foot wounds upregulate mRNAs for efferocytosis pathway genes and display altered efferocytosis signaling via the receptor *Axl* and its ligand *Gas6*. During early inflammation in mouse wounds, we detect upregulation of Axl in dendritic cells and fibroblasts via TLR3-independent mechanisms. Inhibition studies in vivo in mice reveal that Axl signaling is required for wound repair but is dispensable for efferocytosis. By contrast, inhibition of another efferocytosis receptor, Timd4, in mouse wounds decreases efferocytosis and abrogates wound repair. These data highlight the distinct mechanisms by which apoptotic cell detection coordinates tissue repair and provides potential therapeutic targets for chronic wounds in diabetic patients.

## Editor's evaluation

The manuscript reports important new information on how dead cells are cleared from the wound site in order to promote effective repair of the damaged tissue. Interestingly, the authors find that the components of this clearance pathway are abnormally high in diabetics who have difficulty healing wounds and their results suggest that tampering them down may be a therapy to restore normal wound healing.

## Introduction

Proper tissue function and homeostasis require efficient and effective repair of injury. Repair of mammalian tissues requires highly dynamic changes in cellular heterogeneity and communication to correctly heal tissue, usually resulting in a scar rather than true tissue regeneration. Cell death is a common event during tissue injury, and several studies from hydra to mice have shown the importance of apoptosis in the initiation of inflammation to drive reparative processes (*Greenhalgh, 1998*).

**eLife digest** Our skin is constantly exposed to potential damage from the outside world, and it is vital that any injuries are repaired quickly and effectively. Diabetes and many other health conditions can hamper wound healing, resulting in chronic wounds that are both painful and at risk of becoming infected, which can lead to serious illness and death of patients.

After an injury to the skin, the wound becomes inflamed as immune cells rush to the site of injury to fight off infection and clear the wound of dead cells and debris. Some of these dead cells will have died by a highly controlled process known as apoptosis. These so-called apoptotic cells display signals on their surface that nearby healthy cells recognize. This triggers the healthy cells to eat the apoptotic cells to remove them from the wound. Previous studies have linked changes in cell death and the removal of dead cells to chronic wounds in patients with diabetes, but it remains unclear how removing dead cells from the wound affects healing.

Justynski et al. used a genetic technique called single-cell RNA sequencing to study the patterns of gene activity in mouse skin cells shortly after a wound. The experiments found that, as the area around the wound started to become inflamed, the wounded cells produced signals of apoptosis that in turn triggered nearby healthy cells to remove them. Other signals relating to the removal of dead cells were also widespread in the mouse wounds and treating the wounds with drugs that inhibit these signals resulted in multiple defects in the healing process.

Further experiments used the same approach to study samples of tissue taken from foot wounds in human patients with or without diabetes. This revealed that several genes involved in the removal of dead cells were more highly expressed in the wounds of diabetic patients than in the wounds of other individuals.

These findings indicate that for wounds to heal properly it is crucial for the body to detect and clear apoptotic cells from the wound site. Further studies building on this work may help to explain why some diabetic patients suffer from chronic wounds and help to develop more effective treatments for them.

Proper initiation and subsequent resolution of inflammation is essential for tissue repair and progression to the proliferation stage of healing, when fibroblasts, blood vessels, and other tissue-specific cells proliferate and migrate, forming new tissue to repair the wound. While several signaling factors have been shown to induce apoptosis in wounds (*Guerin et al., 2021*), the mechanisms by which apoptotic cells are recognized and regulate tissue repair are not well understood.

The skin is an excellent model to define the mechanisms by which apoptotic cells regulate tissue repair. After injury, mammalian skin undergoes stages of repair beginning with inflammation, which removes debris and pathogens. As inflammation regresses, the proliferative phase leads to the coordination of epidermal keratinocytes, fibroblasts, endothelial, and immune cells to reseal the epidermal barrier and generate a reparative scar including new ECM production and revascularization (*Eming et al., 2014*). Apoptosis occurs after skin injury, and phagocytosis of apoptotic cells – or efferocytosis – by macrophages reduces inflammatory signaling and repair in several tissues (*Bosurgi et al., 2017a*; *Peiseler and Kubes, 2019*). Yet, it is unclear how apoptosis controls skin wound healing.

Apoptotic cell death is characterized by cytomorphological alterations, DNA fragmentation, activation of caspases and other regulators, and finally membrane alterations including outer membrane exposure of phosphatidylserine (PtdSer), which allows the recognition of apoptotic cells by cellular receptors on phagocytes (*Elmore, 2007*). The most well-studied receptors that allow phagocytes to bind and phagocytose apoptotic cells include the TAM (Tyro3, Axl, and Mertk) tyrosine kinases and the TIM (T cell immunoglobulin and mucin domain) family of receptors (*Lemke, 2019*). While TIM receptors can directly bind PtdSer, TAM receptors require their ligands growth-arrest-specific 6 (Gas6) and protein S (Pros1) to bind to PtdSer (*Lew et al., 2014*; *Elliott et al., 2017*).

To understand the cellular and molecular mechanisms by which apoptosis regulates skin wound healing, we performed single-cell RNA sequencing (scRNA-seq) on cells from murine wound beds 24 hr and 48 hr after injury. We found that transcriptional alterations in apoptotic pathways occur in this interval in fibroblasts, monocytes/macrophages, neutrophils, and dendritic cells. In addition, inhibition of two efferocytosis receptors, Axl and Timd4, abrogates proper wound repair. The results provide an

atlas of cellular dynamics during the early stages of wound healing and reveal the essential role of the recognition and clearance of apoptotic cells in driving tissue repair after injury.

## Results

### Dramatic transcriptional heterogeneity during early skin inflammation after injury

To assess cellular and molecular heterogeneity of the wound bed during the inflammatory phase, we performed scRNA-seq on cells isolated from 4 mm full-thickness biopsy punches on mouse back skin at 24 hr and 48 hr after injury (*Shook et al., 2016*). To ensure that we also captured the immediately adjacent tissue as well as cells that may have migrated into the wound site, we used a 6 mm biopsy punch to collect the tissue before isolating cells using enzymatic digestion (*Figure 1A*). By training a neural network to identify cell types based on expression of established marker genes (as in *Wasko et al., 2022*; *Figure 1—figure supplement 1A–B*), we classified four major cell types in the scRNA-seq data, including monocytes/macrophages (Mono/MO), neutrophils (Neut), dendritic cells (DC), and fibroblasts (FB) at both timepoints (*Figure 1B–E*).

Surprisingly, the 24 hr and 48 hr samples clustered separately with minimal overlap, suggesting that dramatic changes occurred in the first 2 days of inflammation (*Figure 1B–D*). The absolute number of macrophages and neutrophils increased >2-fold over this interval (*Figure 1—figure supplement 1C*). While approximately 70% of each of the three immune cell populations was collected at 48 hr, the majority of fibroblasts were found in the 24 hr population (*Figure 1—figure supplement 1D*). We also found that genes upregulated by each cell type were markedly different between timepoints (*Figure 1—figure supplement 1E*), indicating major changes to expression patterns in the same cell type over time.

To explore the changes in the major cell types in early wound beds (fibroblasts, neutrophils, DCs, and monocytes/macrophages), we analyzed the genes that were significantly upregulated in each group relative to the mRNAs expressed by the full dataset to determine gene ontology (GO) terms that were enriched for each cell type (*Figure 1F*). Monocytes/macrophages, neutrophils, and DCs upregulated mRNAs involved in their specific function in inflammation for cytokine production, chemotaxis, and antigen presentation, respectively. Similarly, fibroblasts uniquely upregulated genes involved in extracellular matrix (ECM) organization. Interestingly, DCs, fibroblasts, and neutrophils upregulated genes involved in apoptosis and cell death (*Figure 1F and G*). Neutrophils upregulated the largest number of mRNAs for apoptosis including *Casp3*, which is cleaved to activate apoptosis, and *S100a8* and *S100a9*, which induce apoptosis of several cell types (*Yui et al., 2003*). Fibroblasts and DCs also upregulated apoptotic genes ranging from receptors (*Fgfr1* and *Bmpr1a*), cytokines (*Il12b*), to mediators of apoptotic pathways (*Bcl2l14* and *Cth*).

Since apoptosis also involves post-transcriptional activation of several proteins, we sought to confirm that apoptosis occurred during early skin wound inflammation in vivo. Sections of mouse wounds at 24 hr (*Figure 1H*) and 48 hr (*Figure 1—figure supplement 1F*) were stained with antibodies (Abs) against cleaved (active) caspase 3 to detect activation of apoptosis. We found that significantly more cleaved caspase 3$^+$ cells were present in 48 hr wounds (*Figure 1—figure supplement 1F*). We also detected cleaved caspase 3 in CD11c$^+$ dendritic cells (*Figure 1I*) and Pdgfra$^+$ fibroblasts (*Figure 1J*) in 24 hr wound beds. Overall, these data suggest that apoptosis occurs during inflammation after skin injury.

### Apoptosis recognition receptors, ligands, and downstream factors are expressed in the wound bed

Given the upregulation of mRNAs associated with apoptosis and the relatively low level of apoptotic cells in early wound beds, we hypothesized that efficient efferocytosis clears apoptotic cells during inflammation. Initially, we inspected mRNA levels of efferocytosis receptors, their ligands, and downstream factors in scRNA-seq data from 24 hr and 48 hr wounds (*Figure 2A*). Interestingly, fibroblasts upregulated several genes encoding efferocytosis receptors (*Tyro3*, *Axl*, *Itgav*, and *Slc7a11)*, as well as genes encoding for ligands *Gas6*, *Pros1*, *C3*, *C4b*, and *Mfge8*. DCs, macrophages, and neutrophils also upregulated several receptors that mediate detection of apoptotic

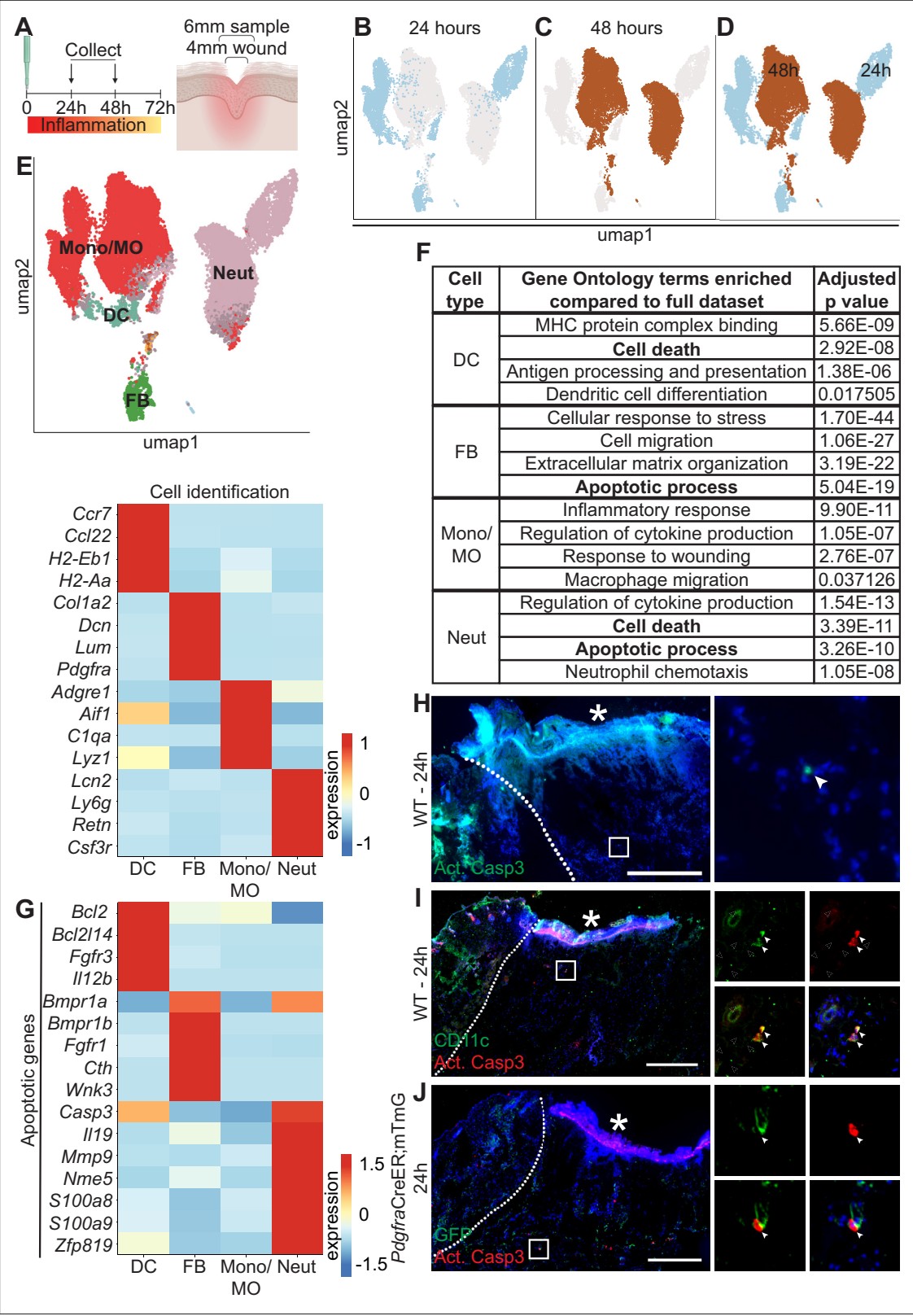

**Figure 1.** Dynamic transcriptional heterogeneity and apoptosis are observed in murine wound beds 24 hr and 48 hr after injury. (**A**) Schematic of experimental design. (**B**) UMAP plot of single-cell RNA sequencing (scRNA-seq) data for cells from 24 hr wound beds in murine back skin. (**C**) UMAP plot of scRNA-seq data for cells from 48 hr wound beds in murine back skin. (**D**) UMAP plot of scRNA-seq data for cells from both 24 hr and 48 hr wound beds in murine back skin annotated by timepoint. (**E**) Top: UMAP plot of scRNA-seq data for cells from 24 hr and 48 hr wound beds annotated by cell

*Figure 1 continued on next page*

*Figure 1 continued*

identity. Bottom: Heatmap of differentially expressed marker genes in 24 hr and 48 hr wound beds. (**F**) Gene ontology terms enriched in each cell type compared to the full dataset. (**G**) Heatmap of differentially expressed apoptosis-related genes from 24 hr and 48 hr wound beds. (**H**) Immunostaining for cleaved caspase 3 (Act. Casp3) (green) in wound bed 24 hr after injury. Arrow indicates cleaved caspase 3[+] cells. (**I**) Immunostaining for CD11c (green) and cleaved caspase 3 (red) in wound bed 24 hr after injury. Arrows indicate double-positive CD11c[+] cleaved caspase 3[+] cells. (**J**) Immunostaining for GFP (green) and cleaved caspase 3 (red) in *Pdgfra*CreER;mTmG wound bed 24 hr after injury. Arrow indicates double-positive GFP[+] cleaved caspase 3[+] cells. * indicates scab. Scale bars = 500 μm. In E and G, expression indicates scaled log-normalized mRNA counts.

The online version of this article includes the following figure supplement(s) for figure 1:

**Figure supplement 1.** Dynamic cellular heterogeneity and evidence of apoptosis is observed in 24 hr and 48 hr wound beds by single-cell RNA sequencing (scRNA-seq).

cells. Macrophages were enriched for the downstream activators *Arg1* and *Retnla*, whereas other cell types upregulated *Socs1* and *Socs3*, which are downstream of the TAM receptors (***Rothlin et al., 2007***).

We noted that Axl's ligand, *Gas6*, and several other genes involved in efferocytosis were expressed predominantly in the fibroblast cluster, but were also lowly expressed in the monocyte/macrophage cluster (***Figure 2A***). Examining the UMAP plot to determine the heterogeneity of efferocytosis gene expression within individual cell types, we observed that *Gas6* was highly expressed by a specific subset of monocyte/macrophage cells. These cells also overexpressed the resident macrophage marker *Lyve1*, the apoptosis receptor *Timd4*, and *Retnla*, a downstream factor of efferocytosis (***Figure 2B***), indicating that they may play a role in apoptosis detection and response in wound healing. *Lyve1* has been identified as a marker for resident macrophages, which are distinct from the majority of wound macrophages that differentiate from bone marrow-derived monocytes and are recruited to the wound after injury (***Lim et al., 2018***; ***Wang et al., 2020***). To confirm the presence of Lyve1[+] resident cells in vivo, we used a *Lyz2*CreER;mTmG mouse model, in which myeloid cells in the bone marrow can be induced to express GFP prior to injury, such that any GFP[+] cells observed in the wound bed are interpreted as newly recruited to the site of injury, while GFP[-] cells are interpreted to be resident to the skin. We observed that Lyve1 was expressed at the protein level both in wound beds (***Figure 2C***) and adjacent to the wound (***Figure 2—figure supplement 1A***) with immunofluorescence staining, and confirmed that these cells were resident rather than recruited to the wound bed after injury, since they did not express GFP. Further, we confirmed that Lyve1[+] cells co-expressed Gas6 protein (***Figure 2D***) and that Timd4 protein (***Figure 2E***) was expressed in wound beds.

The TAM receptor *Axl* was uniquely expressed in the single-cell dataset by both dendritic cells and fibroblasts (***Figure 2A***). Using immunofluorescence staining, we confirmed Axl expression in 24 hr wound beds in both dendritic cells (CD11c) (***Figure 2F***) and fibroblasts (Pdgfra-GFP) (***Figure 2G***), which displayed elongated GFP[+] processes with central localization of Axl. Thus, while we cannot rule out Axl expression by other cell types in skin wounds, these data are consistent with our single-cell results, confirming that Axl is expressed by fibroblasts and dendritic cells during the inflammatory response of skin repair.

We also analyzed mRNA expression of apoptosis-related genes by qPCR to determine their expression in wound beds compared to naive skin (***Figure 2—figure supplement 1B***). All efferocytosis receptor mRNAs studied, including the TAM receptors, were significantly upregulated in the wound bed compared to naive skin, while the ligand *Gas6* was significantly downregulated. The downstream factor of efferocytosis *Retnla* was also significantly downregulated in the wound bed, while *Socs1* and *Socs3* were significantly upregulated.

To analyze efferocytosis activity after injury, we intradermally injected labeled apoptotic neutrophils from bone marrow into wound beds 1, 3, and 5 days after injury. Wound beds were collected 1 hr after injections, sectioned, and imaged to quantify labeled cells (***Figure 2H***). We detected and quantified labeled intact neutrophils and efferocytosed neutrophils in wounds at all three timepoints (***Figure 2H–M***). While wounds at 5 days had significantly more cells undergoing efferocytosis (***Figure 2L***), the efferocytosis rate was generally constant, with a slight but nonsignificant increase at 3 days (***Figure 2J***). Taken together, these data indicate that the machinery for multiple efferocytosis pathways are active in the inflammatory and proliferative stages of wound healing.

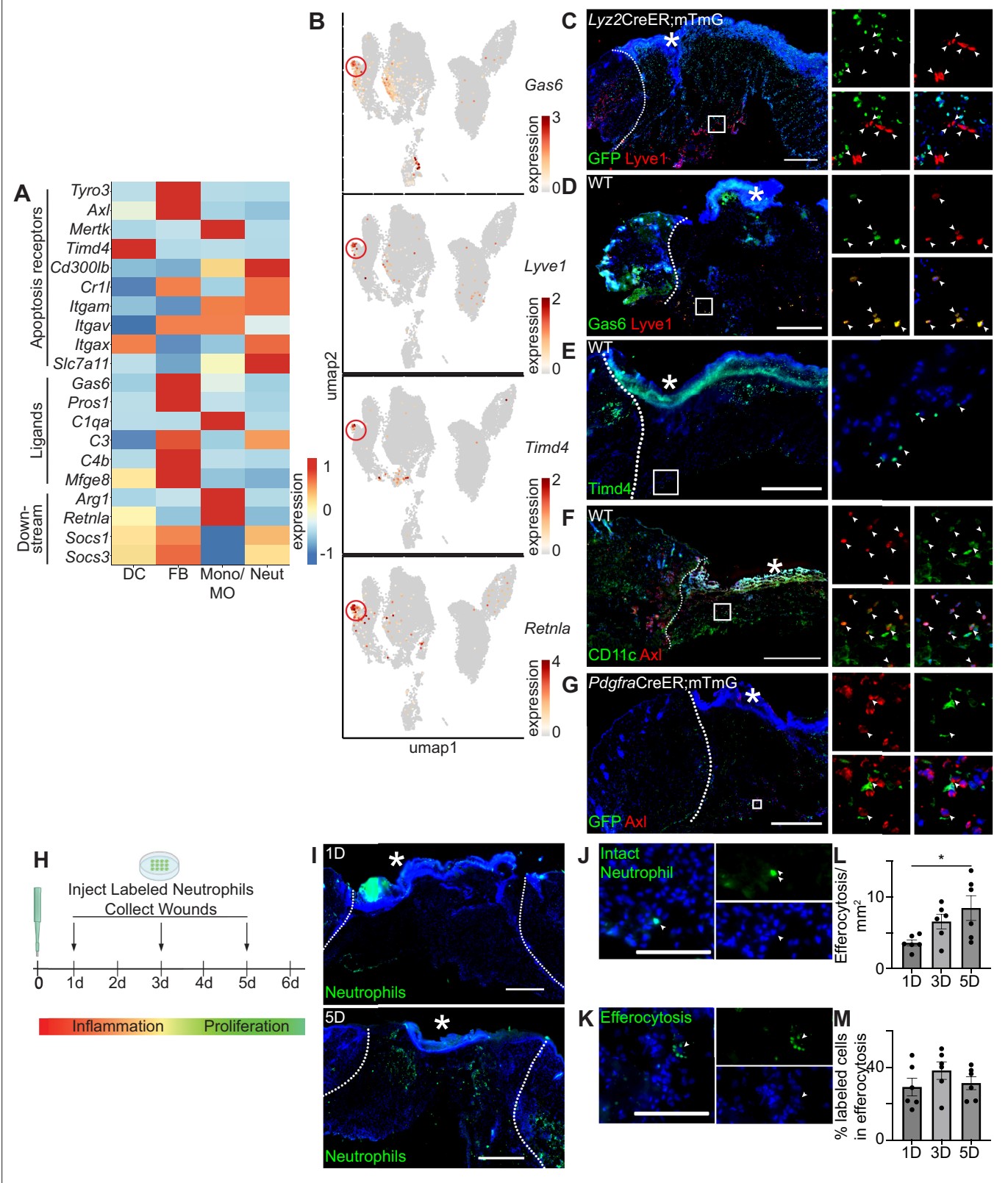

**Figure 2.** Apoptosis detection genes are highly expressed in the wound bed. (**A**) Heatmap of differentially expressed efferocytosis pathway genes in 24 hr and 48 hr wound beds. Expression indicates scaled log-normalized mRNA counts. (**B**) Feature plots showing expression of *Gas6*, *Lyve1*, *Timd4*, and *Retnla* with *Lyve1*+ region highlighted. (**C**) Immunostaining for GFP (green) and Lyve1 (red) in *Lyz2*CreER;mTmG wound bed 24 hr after injury. Arrows indicate Lyve1+ cells. (**D**) Immunostaining for Gas6 (green) and Lyve1 (red) in wild-type (WT) wound bed 24 hr after injury. Arrows indicate double-positive

*Figure 2 continued on next page*

*Figure 2 continued*

Gas6⁺ Lyve1⁺ cells. (**E**) Immunostaining for Timd4 (green) in WT wound bed 24 hr after injury. Arrows indicate Timd4⁺ cells. (**F**) Immunostaining for CD11c (green) and Axl (red) in wound bed 24 hr after injury. Arrows indicate double-positive CD11c⁺ Axl⁺ cells. (**G**) Immunostaining for GFP (green) and Axl (red) in *Pdgfra*CreER;mTmG wound bed 24 hr after injury. Arrows indicate double-positive GFP⁺ Axl⁺ cells. (**H**) Schematic of experimental design. (**I**) Immunostaining for CFSE-stained apoptotic neutrophils in wound beds 1 and 5 days after injury. (**J**) Example of CFSE-stained whole neutrophil in the wound bed. Scale bar = 100 μm. (**K**) Example of active efferocytosis of CFSE-stained whole neutrophil in the wound bed. Scale bar = 100 μm. (**L**) Quantification of instances of efferocytosis observed per mm² in the wound bed. (**M**) Quantification of percentage of all stained cells that are undergoing efferocytosis in the wound bed. In L and M, n=6, error bars indicate mean ± SEM, unpaired t-test, *p<0.05, **p<0.01. * indicates scab. In C–I, scale bars = 500 μm.

The online version of this article includes the following figure supplement(s) for figure 2:

**Figure supplement 1.** Lyve1 and apoptosis receptors, ligands, and downstream factors are expressed in 24 hr and 48 hr wound beds.

## Cell death signaling in human diabetic and non-diabetic wounds

We next set out to explore whether these apoptosis and efferocytosis-related pathways were also relevant in pathological states associated with dysregulated wound healing, such as diabetes. Previous studies have indicated that apoptosis is increased in diabetic wounds, including an elevation in apoptotic lymphocytes (*Arya et al., 2014*). Additional studies have shown that hypoxic environments (such as those found in diabetic wounds) increase macrophage efferocytosis (*Wang et al., 2023*), though other studies have indicated that macrophage efferocytosis is defective in diabetic wounds (*Khanna et al., 2010*).

To analyze the expression of transcriptional changes in apoptosis and efferocytosis signaling in foot wounds, we analyzed genes in these categories in four non-diabetic and five diabetic human patients (to be published elsewhere). We found several alterations in the expression of mRNAs associated with apoptotic signaling pathways between non-diabetic and diabetic foot ulcers (*Figure 3—figure supplement 1A*). Specifically, monocytes/macrophages increased expression of several apoptosis genes (including *BCL2*, *FGFR3*, *FGFR1*, *CTH*, *CASP3*, *IL19*, *NME5*, and *S100A8/9*) in diabetic foot ulcers compared to monocytes/macrophages in non-diabetic wounds.

When we compared the expression of genes associated with the efferocytosis pathway between non-diabetic and diabetic foot ulcers, we found a striking increase in overall expression of efferocytosis pathway genes in diabetic wounds compared to non-diabetic wounds (*Figure 3A*). In particular, the expression of *AXL* increased in all cell types in diabetic wounds, while expression of its ligand *GAS6* increased in all cell types with the exception of basal keratinocytes and mast cells.

To further investigate the *AXL/GAS6* signaling pathway, we analyzed these data with CellChat, which quantitatively infers intercellular signaling networks from scRNA-seq data (*Jin et al., 2021*). This analysis revealed altered signaling between *AXL* and its ligand *GAS6* between the two groups. In non-diabetic patient wounds, *GAS6* expression by several cell types, including monocytes/macrophages and fibroblasts, stimulated signaling via *AXL* receptors on fibroblasts and pericytes. However, in diabetic patient wounds, *GAS6* and *AXL* expression was more robust in several cell types. Notably, monocytes/macrophages increased *GAS6* stimulation and newly expressed *AXL* to receive pathway signaling in diabetic wounds (*Figure 3B*).

To validate these findings, we stained sections of non-diabetic foot wounds and diabetic foot ulcers with Abs against Gas6 and CD68, a macrophage marker. Indeed, we found that Gas6 staining was more prevalent in diabetic skin sections, and that it colocalized with the macrophage marker (*Figure 3C*). Further, when quantified via corrected total fluorescence (CTF), more Gas6 staining was observed in diabetic samples, though this was not significant. This supported the CellChat results, indicating that macrophages express Gas6 more highly in diabetic wounds. Taken together, these data indicate that diabetic wounds may activate and modulate *GAS6-AXL* signaling, and suggest a potential avenue for future research.

## TLR3 stimulation is sufficient, but not necessary, for *Axl* upregulation in skin

Next, we sought to examine the molecular mechanisms that induce *Axl* mRNA expression after injury. Prior work showed that *Axl* expression was induced by toll-like receptor 3 (TLR3) stimulation (*Rothlin et al., 2007*) and that TLR3 is essential for skin wound repair (*Lin et al., 2011*). Thus, we experimentally tested the role of TLR3 signaling in *Axl* expression in the skin. scRNA-seq of early wounds

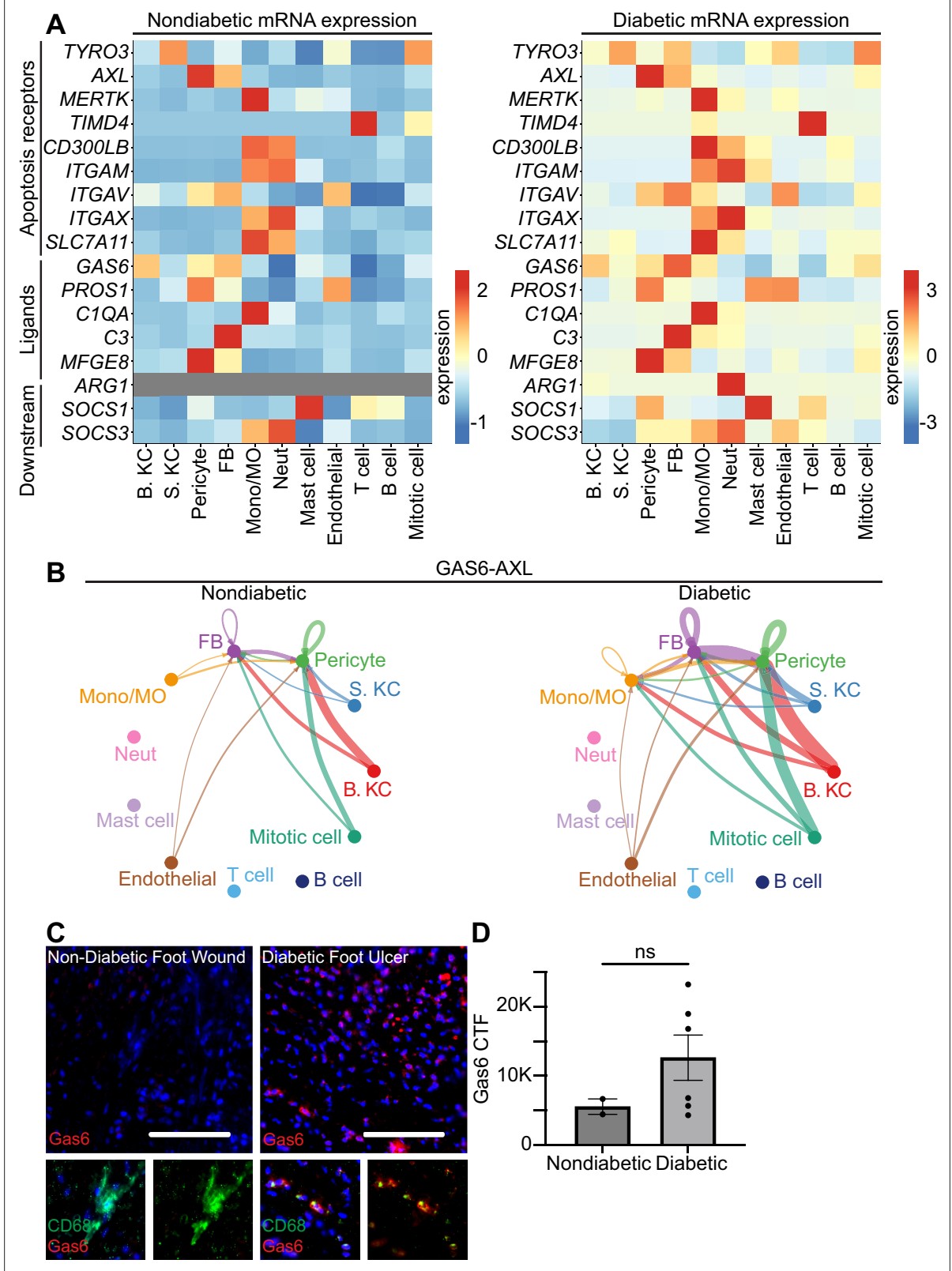

**Figure 3.** Human diabetic wounds have increased efferocytosis signaling expression compared to non-diabetic wounds. (**A**) Heatmaps of differentially expressed genes related to efferocytosis in non-diabetic and diabetic wound beds. Expression indicates scaled log-normalized mRNA counts. (**B**) CellChat circle plot diagrams showing *GAS6-AXL* communication in non-diabetic and diabetic wound beds. (**C**) Left: Immunostaining for CD68 (green) and Gas6 (red) in non-diabetic foot wound. Right: Immunostaining for CD68 (green) and Gas6 (red) in diabetic foot ulcer. Scale bars = 50 μm.

*Figure 3 continued on next page*

*Figure 3 continued*

(**D**) Quantification of Gas6 corrected total fluorescence. Error bars indicate mean ± SEM. n=2 for Non-diabetic, and 6 for diabetic foot wound; ns, nonsignificant.

The online version of this article includes the following figure supplement(s) for figure 3:

**Figure supplement 1.** Human diabetic wounds have altered apoptosis gene expression compared to non-diabetic wounds.

revealed that *TLR* mRNAs are highly expressed in neutrophils and macrophages with a few expressed in dendritic cells and fibroblasts (*Figure 4A*). Interestingly, *TLR3* is unique among the TLRs in that it is expressed by both dendritic cells and fibroblasts, which also express high levels of *Axl* in the single-cell dataset (*Figure 2A*). To determine if TLR3 stimulation upregulates *Axl* expression in the skin, we injected the synthetic double-stranded RNA polyinosinic:polycytidylic acid (poly(I:C)) or a PBS control in naive mouse back skin of either wild-type (WT) or TLR3 knockout (KO) mice. We collected the injection site and surrounding area after 2 hr and processed the skin samples for qPCR and immunostaining (*Figure 4B*). We first analyzed cytokine mRNA expression, a target of TLR3 signaling that promotes inflammation (*Rothlin et al., 2007*). While several inflammatory cytokines were not induced in skin injected with poly(I:C) (*Figure 4—figure supplement 1A*), interferon (IFN)-β (*Ifnb*) was upregulated in WT mice injected with poly(I:C) but not in the skin of *TLR3* KO mice (*Figure 4C*), confirming the specificity of poly(I:C) for activation of TLR3 in the skin (*Alexopoulou et al., 2001*). Axl protein (*Figure 4D*) and mRNA (*Figure 4E*) were induced in naive skin upon poly(I:C) injection, and *Axl* mRNA induction and protein expression were abrogated in skin of *TLR3* KO mice (*Figure 4E* and *Figure 4—figure supplement 1B*). Axl protein expression around the dermal injection site was quantified via CTF (*Figure 4F*), confirming these results. Taken together, these data indicate that TLR3 signaling is sufficient to activate *Axl* expression in naive skin.

Since *Ifnb* was elevated by TLR3 signaling in naive skin, we determined whether injecting recombinant IFN-β intradermally into naive mouse back skin was sufficient to induce Axl protein expression (*Figure 4G*). Two hours after injection, Axl protein was detected by immunostaining in skin injected with IFN-β but not control skin (*Figure 4H and F*). Next, to determine if TLR3 activation was necessary for Axl expression in skin wounds, we analyzed Axl expression in wound beds of *TLR3* KO mice. In contrast to our previous results in naive skin, Axl protein expression was stimulated in *TLR3* KO wound beds similar to WT mice (*Figure 4I*). Thus, these data suggest that while TLR3 is sufficient to drive Axl expression in naive skin, additional mechanisms drive Axl upregulation within skin wounds in its absence.

## Axl is required for skin wound healing

Based on our data showing upregulation of Axl mRNA and protein expression in wound beds, even in the absence of TLR3, we hypothesized that Axl may play a role in wound healing. To examine whether Axl signaling was required for wound repair, we intraperitoneally injected mice 3 hr prior to injury with either a control IgG Ab or anti-Axl function blocking Ab, which binds to Axl's extracellular domain and blocks Axl-mediated viral infection (*Retallack et al., 2016*), and has been shown to inhibit Axl activity in vitro (*Bauer et al., 2012*; *Figure 5A*). Since Axl activity upregulates *Axl* mRNA expression in a positive feedback loop (*Brand et al., 2014*), we analyzed *Axl* mRNA expression in wounds of Ab-treated mice. While we saw a nonsignificant downregulation of *Axl* mRNA at 1 day (D) post injury, *Axl* mRNA was significantly downregulated in skin wounds treated with anti-Axl Ab on day 5 compared to control IgG Ab injected wounds (*Figure 5B*), indicating that the anti-Axl Ab treatment reduced Axl signaling in skin wounds.

Next, we examined the effect of Axl inhibition on apoptotic cell clearance in skin wounds. TUNEL[+] cells were present but rare in both anti-Axl Ab and IgG Ab-treated 1-day wounds (*Figure 5—figure supplement 1A*). Additional staining in sections from wounds 3 and 5 days after injury also showed no significant difference in cleaved (active) caspase 3[+] cells in anti-Axl Ab and IgG Ab-treated wounds (*Figure 5—figure supplement 1B*). Further, we detected no significant changes to gene expression of selected members of the efferocytosis signaling pathway or common inflammatory cytokines 1 day after injury, indicating no major inflammatory defects (*Figure 5—figure supplement 1C*). Thus, additional efferocytosis mechanisms, potentially including other members of the TAM family, likely clear apoptotic cells in early skin wounds despite Axl inhibition.

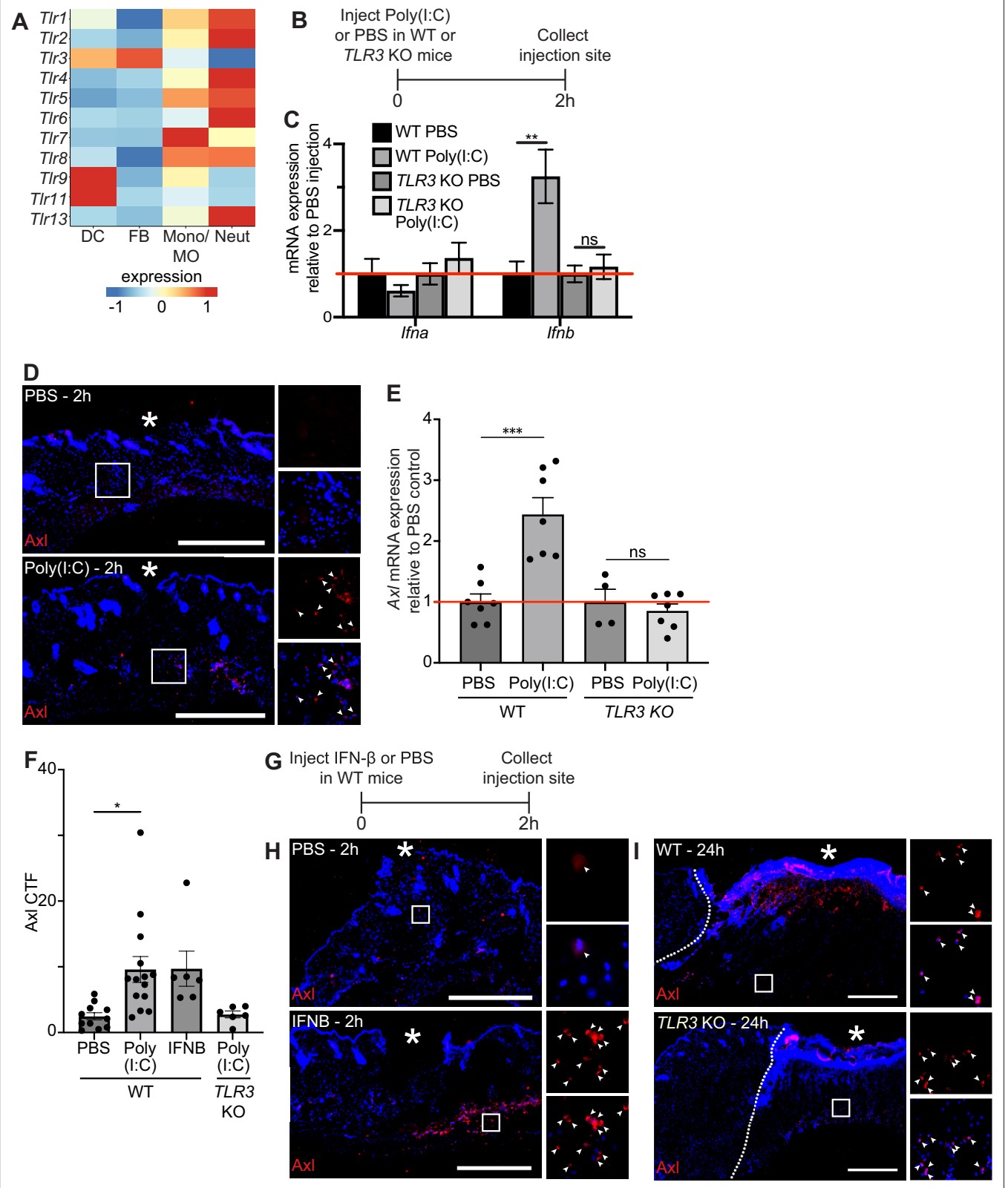

**Figure 4.** TLR3 signaling is sufficient to upregulate Axl in naive skin, but TLR3 is not required for Axl expression in the wound bed. (**A**) Heatmap of differentially expressed TLR genes in 24 hr and 48 hr wound beds. Expression indicates scaled log-normalized mRNA counts. (**B**) Schematic of injection and collection protocol. (**C**) mRNA expression of interferon genes relative to respective PBS injection. Red line indicates normalized control mRNA levels. Error bars indicate mean ± SEM, unpaired t-test, n=4-6 mice; **p<0.01. ns, nonsignificant. (**D**) Immunostaining for Axl (red) in naive back skin

*Figure 4 continued on next page*

*Figure 4 continued*

injected with PBS or Poly(I:C). Arrows indicate Axl⁺ cells. * indicates injection site. (**E**) mRNA expression of *Axl* relative to respective PBS injection control 2 hr after injection. Red line indicates normalized control mRNA levels. Error bars indicate mean ± SEM, unpaired t-test, n=4-8 mice; ***p<0.001. ns, nonsignificant. (**F**) Quantification of corrected total fluorescence for Axl immunostaining in a 1 mm square containing the injection site. n=6-10 mice; Error bars indicate mean ± SEM, one-way ANOVA with multiple comparisons, *p<0.05. (**G**) Schematic of injection and collection protocol. (**H**) Immunostaining for Axl (red) in naive back skin injected with PBS or IFNB. Arrows indicate Axl⁺ cells. * indicates injection site. (**I**) Immunostaining for Axl (red) in wound beds 24 hr after injury in wild-type (WT) or *TLR3* knockout (KO) mice. Arrows indicate Axl⁺ cells. * indicates scab. Scale bars = 500 µm.

The online version of this article includes the following figure supplement(s) for figure 4:

**Figure supplement 1.** Poly(I:C) injection does not elicit Axl expression in *TLR3* knockout (KO) model.

To characterize the impact of Axl inhibition on inflammation at 3 days after injury, we used flow cytometry to quantify the immune cells present in the wound bed (*Figure 5C–D* and *Figure 5—figure supplement 1D*). Interestingly, no significant change was observed in the proportion of dendritic cells, neutrophils, or macrophages that were present in IgG or anti-Axl Ab-treated wound beds. However, the proportion of cell types expressing Axl was altered, with more Axl⁺ macrophages and fewer Axl⁺ DCs present in the anti-Axl Ab-treated group compared to the IgG Ab control.

Despite the relatively normal inflammatory cell numbers at day 3 after injury, wounds treated with anti-Axl Ab exhibited observable healing defects at 5 days post injury compared to wounds treated with control IgG Ab, including a qualitative lack of granulation tissue visible with hematoxylin and eosin (H&E) staining (*Figure 5E*). We also observed that upon Axl inhibition, fibroblast repopulation was significantly reduced (*Figure 5E–F*). Revascularization was also defective; Axl inhibition significantly reduced the CD31⁺ area of wound beds (*Figure 5E–F*). However, we observed no significant difference in the percentage of wound closure by keratinocytes as indicated by ITGA6 staining between the two treatments (*Figure 5—figure supplement 1E*).

We observed significant changes in gene expression at 5 days after injury in wounds lacking Axl activity, including a significant downregulation of *Arg1*, *Tgfb*, and *Il1a*, and significant upregulation of *Ifnb*, *Socs1*, and *Socs3* upon Axl inhibition (*Figure 5—figure supplement 1C*). Taken together, these changes suggest that Axl inhibition impairs proper healing, resulting in significant defects to revascularization and fibroblast repopulation.

## Timd4 function is required for normal skin repair

To further examine the role of efferocytosis receptors in skin wound repair, we abrogated the function of a member of the TIM family of receptors, Timd4. Intraperitoneal injection of a function blocking anti-Timd4 Ab effectively blocks efferocytosis in an atherosclerosis mouse model (*Foks et al., 2016*). Strikingly, we found that wounds of mice treated with anti-Timd4 Ab displayed significantly more TUNEL⁺ apoptotic cells in 1- and 5-day wound beds compared to wounds of IgG Ab-treated mice (*Figure 6A–B*). Similarly, we observed an increase in cleaved caspase 3⁺ cells in wounds of anti-Timd4 Ab-treated mice (*Figure 6—figure supplement 1A*). While this does not preclude the possibility of an increase in apoptosis upon Ab treatment, it is consistent with a defect to efferocytosis.

Inhibition of Timd4 resulted in qualitative defects in granulation tissue in H&E-stained sections of 5-day wounds (*Figure 6C*). Immunostaining of wound sections with Abs against aSMA (*Figure 6D and F*) and ITGA6 (*Figure 6—figure supplement 1B–C*) did not indicate significant changes to either fibroblast repopulation or re-epithelialization, respectively. However, staining for CD31 (*Figure 6E–F*) revealed a significant defect to revascularization in wounds of anti-Timd4 Ab mice compared to wounds of IgG Ab-treated mice, which indicates a defect to proper healing in the proliferative phase of wound repair.

Next, we examined mRNA expression profiles of inflammatory and efferocytosis signaling pathways in IgG Ab and anti-Timd4 Ab-treated wound beds (*Figure 6G*). The inflammatory cytokines *Ifng* and *Ifnb* were significantly upregulated at day 1 or 3, respectively, in Timd4-inhibited wounds compared to control wounds, suggesting altered inflammatory signaling. Interestingly, at day 1 and/ or day 3, *Socs1* and *Socs3* were also significantly upregulated, similar to the gene expression pattern observed after Axl inhibition (*Figure 5I*). Taken together, these results indicate that Timd4 activity is required for reducing apoptotic cells, inflammation gene expression, and revascularization after injury.

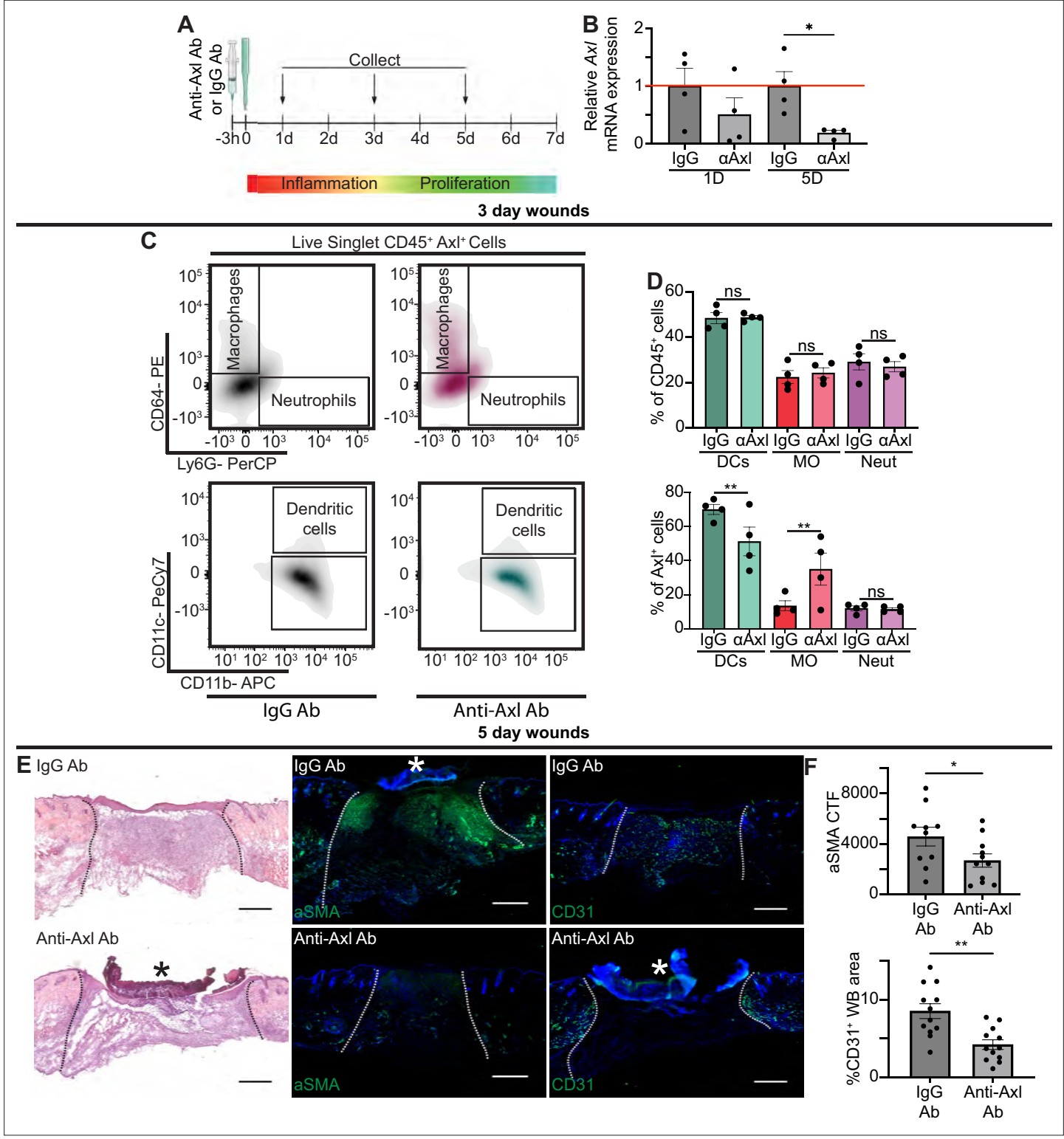

**Figure 5.** Axl antibody (Ab) inhibition results in defects to wound repair and changes to inflammation. (**A**) Schematic of experimental design. (**B**) *Axl* mRNA expression normalized to respective IgG Ab control. Error bars indicate mean ± SEM, one-way ANOVA with multiple comparisons, n=4 mice, *p<0.05. (**C**) Representative flow cytometry gates used to analyze cells isolated from wounds 3 days after injection and injury. Live singlet CD45⁺ cells were identified as macrophages, neutrophils, or dendritic cells via fluorescent antibody staining. (**D**) Top: Quantification of CD45⁺ cells by cell type in anti-Axl Ab or IgG Ab-treated wound beds 3 days after injury. Error bars indicate mean ± SEM, one-way ANOVA with multiple comparisons, n=4 mice, ns, nonsignificant. Bottom: Quantification of Axl⁺ cells by cell type in anti-Axl Ab or IgG Ab-treated wound beds 3 days after injury. Error bars indicate

*Figure 5 continued on next page*

*Figure 5 continued*

mean ± SEM, two-way ANOVA with multiple comparisons, n=4 mice, **p<0.01. ns, nonsignificant. (**E**) Left: Hematoxylin and eosin (H&E) staining of wound beds 5 days after antibody injection and injury. Center: Immunostaining for aSMA (green) in wound beds 5 days after antibody injection and injury. Right: Immunostaining for CD31 (green) in wound beds 5 days after antibody injection and injury. * indicates scab. (**F**) Top: Quantification of aSMA corrected total fluorescence. Error bars indicate mean ± SEM, unpaired t-test, n=10 mice, *p<0.05. Bottom: Quantification of CD31+ pixels in wound bed. Error bars indicate mean ± SEM, unpaired t-test, **p<0.01. Scale bars = 500 μm.

The online version of this article includes the following figure supplement(s) for figure 5:

**Figure supplement 1.** Analysis of anti-Axl antibody (Ab) and IgG control-treated wound beds.

## Discussion

Here, we provide an atlas for the dynamic changes that occur in the early inflammatory stage of wound repair in the skin at the single-cell level. We found that apoptotic and efferocytosis pathways were upregulated in distinct cell types in mouse wounds and in diabetic foot wounds of human patients. Using functional inhibition studies in mouse wounds, our data show that efferocytosis receptors, Axl and Timd4, have differential effects on efferocytosis and wound repair.

We found that apoptotic genes were upregulated in most of the cell types present in early wounds, which likely allows the dynamic shifts in transcriptional profiles of inflammatory cells we observed in the first few days of tissue repair. Apoptotic cells are required for tail regeneration of *Xenopus laevis* (*Tseng et al., 2007*) and tissue regeneration of planaria (*Hwang et al., 2004*) and have been widely implicated in early wound repair (*Greenhalgh, 1998*; *Arya et al., 2014*). Mechanistically, apoptotic cells can release signaling factors including Wnts and PGE2 to promote proliferation of tissue resident cells and tissue repair in multiple species (*Codispoti et al., 2019*; *Li et al., 2010*; *Fuchs et al., 2013*). Detection of apoptotic cells has also been linked to inflammation and context-dependent integration of IL-4 signaling to activate an anti-inflammatory and tissue repair gene program (*Bosurgi et al., 2017a*).

Exposure of PtdSer on the outer leaflet of apoptotic cell membranes acts as a sensor for engulfment in evolutionarily conserved mechanisms (*Rothlin and Ghosh, 2020*). Our data suggest that in skin wounds, apoptotic cells are rapidly detected and removed via efferocytosis, which is consistent with the large number of phagocytic neutrophils and macrophages in early wounds. We also found that mRNAs encoding efferocytosis receptors and signaling pathways are upregulated in the early skin wound beds and in diabetic foot wounds, and confirmed protein upregulation of the efferocytosis receptors Axl and Timd4 in murine wounds. Interestingly, efferocytosis receptors were upregulated in professional phagocytes as well as in fibroblasts, which also increased expression of a large number of efferocytosis ligands in both mouse wounds and diabetic foot ulcers (*Figures 2 and 3*). These data resonate with recent data suggesting that fibroblasts can engulf apoptotic endothelial cells to alter their contractility, migration, and ECM production (*Romana-Souza et al., 2021*), and may indicate a direct role for fibroblasts in modulating the inflammatory milieu of early wounds through detection of apoptotic cells.

Growing evidence suggests that distinct efferocytosis receptors elicit cell- and tissue-specific responses. Axl and Mer exhibit differential ligand specificity and shedding upon activation (*Zagórska et al., 2014*). Furthermore, *Axl* null bone marrow-derived macrophages are unable to perform efferocytosis with TLR3 activation in vitro, which suggests that Axl plays a central role in efferocytosis in inflammatory environments (*Bosurgi et al., 2017b*). Here, we show that in early murine skin wounds, Axl inhibition did not impact efferocytosis in a detectable manner, but rather Axl expression shifted from dendritic cells to macrophages, suggesting possible compensatory mechanisms via other efferocytosis receptors. Furthermore, we did not detect differences in *Axl* expression in *TLR3* null and control wounds, further suggesting the inflammatory wound environment in vivo displays activation of *Axl* receptor expression independent of TLR3. Axl may function in skin wounds by interacting with other tyrosine kinase receptors, which has been shown for EGFR, MET, and PDGFR (*Meyer et al., 2013*). Interestingly, Axl/Gas6 signaling can also regulate tumorigenesis to support tumor cell survival (*Linger et al., 2008*), migration (*Wilson et al., 2014*), and angiogenesis (*Li et al., 2009*; *Zhu et al., 2019*). Future work exploring Axl's function in a cell type-dependent manner, including possibly directly activating angiogenesis in skin wounds, will decipher these possibilities.

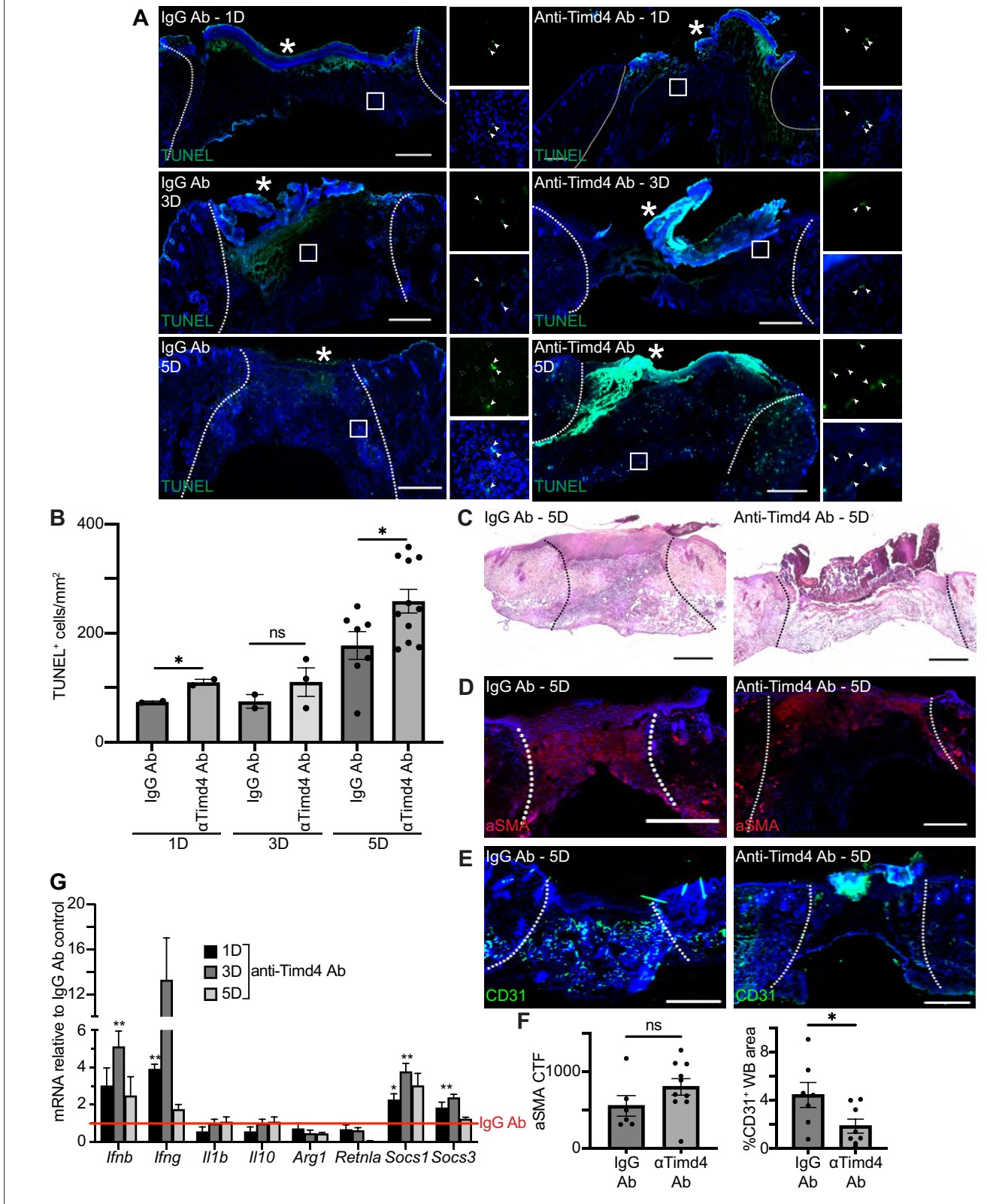

**Figure 6.** Timd4 function is required for efferocytosis during skin repair. (**A**) Immunostaining for TUNEL (green) in wound beds injected with anti-Timd4 antibody (Ab) or IgG Ab control 1, 3, or 5 days after injury. Arrows indicate TUNEL+ cells. * indicates scab. (**B**) Quantification of TUNEL+ cells per mm² in the wound bed. Error bars indicate mean ± SEM, one-way ANOVA, n=3-7 mice; *p<0.05, ns, nonsignificant. (**C**) Hematoxylin and eosin (H&E) staining of wound beds 5 days after antibody injection and injury. (**D**) Immunostaining for aSMA (red) in wound beds 5 days after antibody injection and

*Figure 6 continued on next page*

*Figure 6 continued*

injury. (**E**) Immunostaining for CD31 (green) in wound beds 5 days after antibody injection and injury. (**F**) Left: Quantification of aSMA corrected total fluorescence. Error bars indicate mean ± SEM. ns, nonsignificant. Right: Quantification of CD31$^+$ pixels in wound bed. Error bars indicate mean ± SEM, unpaired t-test, n=6-9 mice; *p<0.05. (**G**) mRNA expression of genes relative to respective IgG Ab control. Red line indicates normalized control mRNA levels. Error bars indicate mean ± SEM, n=4-6 mice; unpaired t-test *p<0.01, **p<0.01. Scale bars = 500 μm.

The online version of this article includes the following figure supplement(s) for figure 6:

**Figure supplement 1.** Re-epithelialization is not significantly altered in anti-Axl antibody and IgG control-treated wound beds.

Our data also implicate an important role for resident macrophages in regulating early wound repair mechanisms. In particular, we identified the expression of several efferocytosis receptors and ligands on Lyve1$^+$ resident macrophages including Timd4. Timd4 is expressed on tissue resident macrophages in multiple tissues including in the peritoneal and resident cardiac macrophages. In the heart, Lyve1$^+$ resident macrophages act in a cardioprotective manner during myocardial infarction (*Dick et al., 2019*) and are immunoregulatory and promote engraftment of cardiac allografts (*Thornley et al., 2014*). Interestingly, similar to our data, Timd4 promotes efferocytosis in the heart. During cardiac infarction, Timd4 is also required for T cell responses and support of regulatory T cells (Tregs) (*Foks et al., 2016*). In the skin, Timd4 is essential for allograft survival (*Yeung et al., 2013*), which has been proposed to act through DCs to support Tregs. Indeed, Timd4 is expressed in DCs in later stages of wound repair (*Haensel et al., 2020*). Our future studies will test the role of Timd4 on DCs and resident macrophages, T cell immunity, and how they impact wound repair phenotypes noted with Timd4 inhibition.

In summary, our data implicate apoptosis recognition receptors as an important regulator of skin wound healing. We find that early murine wound beds and human diabetic foot wounds significantly upregulate apoptotic genes and receptors that recognize apoptotic cells and that inhibition of multiple apoptotic receptors impairs wound repair. Given the importance of apoptosis in wound repair and the expression of distinct apoptotic receptors in different cells, skin wound healing is an excellent model to decipher the mechanisms by which distinct cells recognize and respond to interactions with apoptotic cells. Given further research into the healing phenotypes of efferocytosis-inhibited wounds, it may be that targeting these mechanisms may reveal therapeutic avenues that promote healing in chronic, non-healing wounds, particularly in diabetic patients.

## Methods
### Animals

WT C57BL/6J mice (Strain #:000664), B6;129S1-Tlr3tm1Flv/J (*TLR3* KO) mice (Strain #:005217), Lyz2tm1(cre/ERT2)Grtn/J (*Lyz2*CreER) mice (Strain #:031674), and B6.129(Cg)-Gt(ROSA)26Sortm4(ACTB-tdTomato,-EGFP)Luo/J (mTmG) mice (Strain #:007676) were purchased from The Jackson Laboratories. *Pdgfra*CreER mice were developed in the laboratory of B. Hogan (Duke University, Durham, NC, USA). Mice were maintained through routine breeding in an Association for Assessment and Accreditation of Laboratory Animal Care (AALAC)-accredited animal facility at Yale University (Protocol # 11248). Animals were maintained on a standard chow diet ad libitum (Harlan Laboratories, 2018S) in 12 hr light/dark cycling. Up to five injured mice were housed per cage. All experimental procedures were approved and in accordance with the Institutional Animal Care and Use Committee. For experiments using intraperitoneal (i.p.) tamoxifen administration, 100 μL of 30 mg/mL tamoxifen (Sigma-Aldrich) in sesame oil was injected daily for 3 days prior to experiments.

### Human subjects

Diabetic and non-diabetic adults with chronic foot ulcers that were undergoing skin wound debridement were consented to donate discarded tissue for this study (IRB approval # 1609018360). The diabetic foot ulcer specimens for scRNA-seq were obtained from five individuals diagnosed with Type 2 diabetes. The non-diabetic foot wound specimens for scRNA-seq were obtained from four individuals, two of whom provided multiple samples. The diabetic foot ulcer specimens for immunofluorescence staining were obtained from five individuals diagnosed with Type 2 diabetes and one individual diagnosed with Type 1 diabetes. The non-diabetic foot wound specimens for immunofluorescence

staining were obtained from two individuals. Demographic information listed in *Supplementary file 1* .

## Wounding

Seven- to 9-week-old male mice were wounded during the telogen phase of hair cycling. Mice were anesthetized using isoflurane and six full-thickness wounds, at least 4 mm apart, were made on shaved back skin using a 4 mm biopsy punch (Millitex). Animals were sacrificed at noted intervals after injury and wound beds were processed for subsequent analysis.

## Single-cell digestion of mouse wounds

Wound beds were digested for scRNA-seq analysis in a buffer of Roswell Park Memorial Institute (RPMI) medium with glutamine (Gibco), Liberase Thermolysin Medium (TM) (Roche), DNase, *N*-2-hydroxyethylpiperazine-*N*-2-ethane sulfonic acid (Gibco), sodium pyruvate (Gibco), non-essential amino acids (Gibco), and antibiotic-antimycotic (100X) (Gibco). Blood cells were removed with ammonium-chloride-potassium (ACK) lysing buffer (Gibco). Cells were resuspended in Dulbecco's Modified Eagle's Medium (DMEM) (ATCC) with 0.1% bovine serum albumin (BSA) for analysis.

## Human skin collection and processing for scRNA-seq

Skin wound specimens were collected at the clinical setting in PBS with antibiotic-antimycotic (100X) (Gibco), and transported to the lab on ice for processing. All specimens were processed within 3 hr of collection. The specimen was cleaned by immersion with 10% betadine, 70% ethanol, and PBS. Excess blood and subcutaneous fat were removed, and skin was mechanically minced before being digested in an enzyme cocktail consisting of dispase (Stemcell Technologies), collagenase I (Worthington), and collagenase II (Worthington) in 0.25% trypsin-EDTA (Gibco). Blood cells were removed with ACK lysing buffer (Gibco). Cells were resuspended in DMEM (ATCC) with 0.1% BSA for analysis.

## Human skin collection and processing for staining

Human tissue sections were kept for at least 24 hr in 10% neutral buffered formalin and dehydrated in alcohol diluted in 0.9% NaCl/dH$_2$O; the sequence being 30% EtOH, 50% EtOH, 70%EtOH. Specimens remained in 70% EtOH overnight. The following day, sections were dehydrated further in 85%, 95%, and 100% ethanol diluted in distilled water before being preserved in 50% EtOH/ 50% xylene and then 100% xylene. Tissue sections were allowed to soak in paraffin for at least 1 hr and then embedded in disposable molds with melted paraffin wax. Five μm tissue sections were obtained and adhered to charged slides for histological analysis.

## Single-cell data of mouse samples

scRNA-seq data from 24 hr and 48 hr mouse wound beds were processed using the standard cellranger pipeline (10X Genomics). Downstream analysis was performed using the Scanpy package in Python (*Wolf et al., 2018*). Cells were filtered for quality control to avoid doublets and dead cells. Dimensionality reduction and downstream data visualization were completed using the Scanpy implementation of UMAP (*McInnes et al., 2020*) and the ShinyCell package in R (*Ouyang et al., 2021*), respectively. Data is presented as scaled log-normalized mRNA counts (i.e., expression).

We adapted the cell-type annotation pipeline from *Kumar et al., 2018*, to label our sequencing data by broad cell type, as was done in *Wasko et al., 2022*. Differentially expressed genes (DEGs) across timepoints were calculated using the rank_genes_groups function from the Scanpy module in Python with default parameters. We then performed enrichment analysis on the top (logfoldchanges >1.5) DEGs for each group using g:Profiler (*Raudvere et al., 2019*), with the GO knowledgebase as the reference database.

## Neutrophil injection

Cells were collected from WT bone marrow as previously described (*Rios et al., 2017*). Neutrophils were isolated using MACS Ly6G beads (Miltenyi Biotech 130-120-337). Neutrophils were stained with CFSE cell labeling kit (Abcam ab113853) for 15 min. Stained neutrophils were incubated overnight in RPMI medium to induce apoptosis. 10,000 neutrophils in 50 μL PBS or 50 μL PBS control were

injected under the scab and into the wound bed of 1-, 3-, or 5-day wounded mice. Injected mice were sacrificed after 1 hr.

## Single-cell data of human samples

Human scRNA-seq data from human samples were generated from 10X Genomics 3'-end single-cell gene expression V2. Analysis was performed in Scanpy (*Wolf et al., 2018*). Cells were filtered for doublets and dead cells for downstream analysis. Batch correction was performed to integrate cells across samples using Scanorama (*Hie et al., 2019*). Gene expression was scaled, log-transformed, and normalized. Dimensionality reduction was done using PCA and UMAP in Scanpy (*McInnes et al., 2020*). DEGs were calculated using Wilcoxon rank_genes_groups function from Scanpy. Cell-cell communication analysis and data visualization of circos plot were performed using CellChat (*Jin et al., 2021*). CellChat is superior to similar methods because its algorithm accounts for the roles of both signaling cofactors and protein-protein signaling in its predictions of ligand-receptor interaction (*Bridges and Miller-Jensen, 2022*). CellChat is available as an open-source software package in R.

## Staining and imaging

Mouse skin and wound beds were embedded in optimum cutting temperature compound (VWR) and wound beds were sectioned through their entirety to identify the center. 7 µm or 14 µm cryosections were processed as previously described (*Shook et al., 2016*) and stained with Abs listed below or H&E.

Human tissue slides were warmed in the oven at 60°C for an hour before being deparaffinized and rehydrated in ethanol diluted in water. Antigen retrieval was performed at 80°C for an hour and after slides had cooled, nonspecific blocking was performed, the primary Ab was added, and the slides rested overnight at 4°C. The next day, slides were warmed to room temperature, washed, and the secondary Ab was allowed to penetrate for 1 hr at room temperature. Prolong Gold plus DAPI was added to each tissue section and cover slides were added. Slides rested at 4°C for at least 1 day before imaging.

Composite images were acquired using the tiles module on a Zeiss AxioImager M1 (Zeiss) equipped with an Orca camera (Hamamatsu).

## Quantitative real-time PCR

Whole wound bed samples were digested using TRIzol LS (Invitrogen). RNA was extracted from the aqueous phase using the RNeasy Plus Mini Kit (QIAGEN). cDNA was generated using equal amounts of total RNA with the Superscript III First Strand Synthesis Kit (Invitrogen) per manufacturer's instructions. All quantitative real-time PCR was performed using SYBR green on a LightCycler 480 (Roche). Primers for specific genes are listed below. Results were normalized to β-actin as previously described.

## Injections

Seven- to 9-week-old male mice were injected intradermally with 10 µL PBS, 50 µg/mL Poly(I:C) (Invivogen), or 5 µg/mL IFNB (R&D) with 0.5% BSA. The injection site was isolated using a 6 mm biopsy punch 2 hr after injection and processed for staining.

To inhibit signaling pathways in vivo, 7- to 9-week-old male mice were injected intraperitoneally with 25 µg/100 µL anti-Axl Ab (R&D), 200 µg/100 µL anti-Timd4 (BioXCell), or equivalent unit IgG control (R&D) in PBS 3 hr before wounding.

## Flow cytometry

Mouse wound beds were dissected and digested into single cells using Liberase TM (Roche) and cells were suspended in fluorescence-activated single-cell sorting (FACS) staining buffer (0.05% BSA in DMEM). Digested tissue was filtered with a 70 µm and 40 µm cell strainer prior to centrifugation. Cell suspensions were stained with Abs for 30 min on ice. Dendritic cells were defined as CD11b$^+$ Cd11c$^+$ cells; macrophages were defined as CD11b$^+$ CD11c$^-$ CD64$^+$ Ly6G$^-$ cells; neutrophils were defined as CD11b$^+$ CD11c$^-$ CD64$^-$ Ly6G$^+$ cells. To exclude dead cells, Sytox Blue (Invitrogen, 1:1000) was added immediately before analysis or sorting using a FACS Aria III with FACS DiVA software (BD Biosciences). Flow cytometry analysis was performed using FlowJo Software (FlowJo).

## Image quantification

Histological quantification for each wound bed was conducted on multiple central sections for each wound bed when available. The percentage of the wound bed covered by ITGA6 staining (re-epithelialization) and CTF for Axl (in a 1 mm square around the injection site) or aSMA (in the wound bed) were calculated using ImageJ software (National Institutes of Health, Bethesda, MD, USA) as described previously (*Schmidt and Horsley, 2013*; *Shook et al., 2018*). Revascularization (CD31⁺) was calculated using Adobe Photoshop to measure the total pixels positive for Ab staining divided by the total number of pixels in wound beds. Cell death was quantified using the RETINA Analysis Toolkit in FIJI (https://imagej.net/plugins/retina-analysis-toolkit).

## Statistics

To determine significance between two groups, comparisons were made using Student's t-test. Analyses across multiple groups were made using a one- or two-way ANOVA with Bonferroni's post hoc using GraphPad Prism for Mac (GraphPad Software, La Jolla, CA, USA) with significance set at $p < 0.05$. Sample sizes were determined using power analysis and taking into consideration our experience with the wounding model.

## Antibodies

TUNEL kit: Click-iT Plus TUNEL Assay for In Situ Apoptosis Detection, Alexa Fluor 488 dye Thermo Fisher C10617
Axl: Bioss Cat# bs-5180R, RRID:AB_11110961
Axl: R&D Systems Cat# AF154, RRID:AB_354852
CD31: BD Biosciences Cat# 550274, RRID:AB_393571
CD31: Millipore Cat# MAB1398Z, RRID:AB_94207
CD68: Abcam Cat# ab237968
CD11c: Abcam Cat# ab33483, RRID:AB_726084
Cleaved Caspase 3: Cell Signaling Technology Cat# 9661, RRID:AB_2341188
Gas6: Thermo Fisher Scientific Cat# PA5-103199, RRID:AB_2852567
Gas6: Bioss Cat# bs-7549R-A488
GFP: Abcam Cat# ab13970, RRID:AB_300798
IgG: R&D Systems Cat# AB-108-C, RRID:AB_354267
IgG: Bio X Cell Cat# BE0290, RRID:AB_2687813
ITGA6: R&D Systems Cat# MAB13501, RRID:AB_2128311
Lyve1: Abcam Cat# ab14917, RRID:AB_301509
Timd4: Thermo Fisher Scientific Cat# PA5-116045, RRID:AB_2900679
Timd4: Bio X Cell Cat# BE0171, RRID:AB_2687695
aSMA: Abcam Cat# ab5694, RRID:AB_2223021

## FACS antibodies

Axl: BD Biosciences Cat# 748032, RRID:AB_2872493 - 1:200
Ly6G: BioLegend Cat# 127654, RRID:AB_2616999 - 1:400
CD206: BioLegend Cat# 141710, RRID:AB_10900445 - 1:250
CD11c: BioLegend Cat# 117318, RRID:AB_493568 - 1:500
CD40: BioLegend Cat# 124618, RRID:AB_2075922 - 1:200
CD64: BioLegend Cat# 139304, RRID:AB_10612740 - 1:300
CD45: Thermo Fisher Scientific Cat# 47-0451-82, RRID:AB_1548781 - 1:200
CD11b: BioLegend Cat# 101212, RRID:AB_312795 - 1:200
Sytox Blue: Invitrogen Cat# S34857

## Primers

Arg1 F: CATTGGCTTGCGAGACGTAGAC
Arg1 R: GCTGAAGGTCTCTTCCATCACC

Axl F: CGAGAGGTGACCTTGGAAC
Axl R: AGATGGTGGAGTGGCTGTC
Β-actin F:ATCAAGATCATTGCTCCTCCTGAG
B-actin R: CTGCTTGCTGATCCACATCTG
C3 F: CCAGCTCCCCATTAGCTCTG
C3 R: GCACTTGCCTCTTTAGGAAGTC
C4b F: ACTTCAGCAGCTTAGTCAGGG
C4b R: GTCCTTTGTTTCAGGGGACAG
Cd47 F: TGCGGTTCAGCTCAACTACTG
Cd47 R: GCTTTGCGCCTCCACATTAC
Cd300lb F: TGCAGGGTCCTCATCCGAT
Cd300lb R: TGTCCGTGTCATTTTGCCTGA
C1qa F: ATGGAGACCTCTCAGGGGATG
C1qa R: ATACCAGTCCGGATGCCAGC
Cr1l F: ATGGAGGTCTCTTCTCGGAGT
Cr1l R: GGCCGAAGGCTACAAGGAG
Gas6 F:ATGAAGATCGCGGTAGCTGG
Gas6 R: CCAACTCCTCATGCACCCAT
Gla F: TCTGTGAGCTTGCGCTTTGT
Gla R: GCAGTCAAGGTTGCACATGAAA
Ifna F: CTTCCACAGGATACTGTGTACCT
Ifna R: TTCTGCTCTGACCACCCTCCC
Ifnb F: ATGAGTGGTGGTTGCAGGC
Ifnb R: TGACCTTTCAAATGCAGTAGATTCA
Ifng F: CAGGCCATCAGCAACAACATAAGC
Ifng R: ACCCCGAATCAGCAGCGACTC
Il10 F: GCCCAGAAATCAAGGAGCATT
Il10 R: TGCTCCACTGCCTTGCTCTTA
Il17 F: GTCGAGAAGATGCTGGTGGGTGTG
Il17 R: ACGTGGGGGTTTCTTAGGGGTCAG
Il6 F: AGCCCACCAAGAACGATAGTC
Il6 R: TTGTGAAGTAGGGAAGGCCG
Il1a F: TTGGTTAAATGACCTGCAACA
Il1a R: GAGCGCTCACGAACAGTTG
Il1b F: CTCATTGTGGCTGTGGAGAAG
Il1b R: ACACTAGCAGGTCGTCATCAT
Itgam F: TTCCTGGTGCCAGAAGCTGAA
Itgam R: CCCGTTGGTCGAACTCAGGA
Itgav F: CCGTGGACTTCTTCGAGCC
Itgav R: CTGTTGAATCAAACTCAATGGGC
Itgax F: CCCACCACTTCCTCCTGTAAC
Itgax R: AGCAATTGGGTCACAGGTTC
Mertk F: GGCTTTTGGCGTGACCATG
Mertk R: AGTTCATCCAAGCAGTCCTC
Mfge8 F: AGATGCGGGTATCAGGTGTGA
Mfge8 R: GGGGCTCAGAACATCCGTG
Pros1 F: TGGCAAGGAGACAGGTGTCAGT
Pros1 R: GAGCAGTGGTAACTTCCAGGAG
Retnla F: CCAATCCAGCTAACTATCCCTCC
Retnla R: CCAGTCAACGAGTAAGCACAG
Sirpa F: CCACGGGGAAGGAACTGAAG
Sirpa R: ACGTATTCTCCTGCGAAACTGTA
Slc7a11 F: GGCACCGTCATCGGATCAG
Slc7a11 R: CTCCACAGGCAGACCAGAAAA
Socs1 F: CCGTGGGTCGCGAGAAC

Socs1 R: AACTCAGGTAGTCACGGAGTACCG
Socs3 F: TCCCATGCCGCTCACAG
Socs3 R: ACAGGACCAGTTCCAGGTAATTG
Tgfb F: ACTGTGGAAATCAACGGGATCA
Tgfb R: CTTCCAACCCAGGTCCTTCC
Timd4 F: AGCTTCTCCGTACAGATGGAA
Timd4 R: CCCACTGTCACCTCGATTGG
Tnfa F: TGTCTACTCCTCAGAGCCCC
Tnfa R: TGAGTCCTTGATGGTGGTGC
Tyro3 F: GAGGATGTCCTCATTCCAGAGC
Tyro3 R: CACTGCCACTTTCACGAAGGAG
Vegfa F: CGACACGGGAGACAATGGGATGAA
Vegfa R: AGGGGCGGGGTGCTTTGTAGACT

## Inclusion and diversity

One or more of the authors of this paper self-identifies as: (1) an underrepresented ethnic minority in their field of research or within their geographical location, (2) a gender minority in their field of research, and (3) a member of the LGBTQ+ community. While citing references scientifically relevant for this work, we also actively worked to promote gender balance in our reference list.

## Acknowledgements

We would like to thank members of the Horsley and Miller-Jensen laboratory for their feedback and critical analysis of these data and manuscript. We would also like to thank the Yale Animal Resources Center (YARC) staff for animal husbandry and the Yale Science Building (YSB) Imaging core facility for confocal use. VH is funded by NIH-NIAMS R01s AR076938, AR0695505, AR075412. KM-J is funded by NIH U01-CA238728, R01-CA238728, and R01-GM123011.

## Additional information

### Competing interests

Valerie Horsley: Reviewing editor, eLife. The other authors declare that no competing interests exist.

### Funding

| Funder | Grant reference number | Author |
|---|---|---|
| Gruber Foundation | | Olivia Justynski |
| National Institutes of Health | Predoctoral Training Program in Cellular and Molecular Biology | Olivia Justynski |
| National Institute of Arthritis & Musculoskeletal & Skin Diseases | R01 AR079232 | Olivia Justynski Will Krause Maria Fernanda Forni Teresa Sandoval-Schaefer Kristyn Carter Valerie Horsley |
| National Institute of Arthritis & Musculoskeletal & Skin Diseases | R01 AR075412 | Olivia Justynski Will Krause Maria Fernanda Forni Kristyn Carter Teresa Sandoval-Schaefer Valerie Horsley |

| Funder | Grant reference number | Author |
| --- | --- | --- |
| National Institute of Arthritis & Musculoskeletal & Skin Diseases | R01 AR076938 | Olivia Justynski<br>Will Krause<br>Maria Fernanda Forni<br>Kristyn Carter<br>Teresa Sandoval-Schaefer<br>Valerie Horsley |
| National Institutes of Health | T32 T32GM007223 | Olivia Justynski |
| National Institutes of Health | U01-CA238728 | Kathryn Miller-Jensen |
| National Institutes of Health | R01-CA238728 | Kathryn Miller-Jensen |
| National Institutes of Health | R01-GM123011 | Kathryn Miller-Jensen |

The funders had no role in study design, data collection and interpretation, or the decision to submit the work for publication.

## Author contributions

Olivia Justynski, Conceptualization, Validation, Investigation, Visualization, Writing – original draft, Writing – review and editing; Kate Bridges, Conceptualization, Formal analysis, Visualization, Writing – original draft; Will Krause, Maria Fernanda Forni, Kristyn Carter, Validation, Investigation; Quan M Phan, Diane E King, Formal analysis, Investigation; Teresa Sandoval-Schaefer, Henry C Hsia, Michael I Gazes, Steven D Vyce, Investigation; Ryan R Driskell, Supervision; Kathryn Miller-Jensen, Conceptualization, Funding acquisition, Visualization, Writing – original draft, Writing – review and editing; Valerie Horsley, Conceptualization, Supervision, Funding acquisition, Visualization, Writing – original draft, Writing – review and editing

## Author ORCIDs

Olivia Justynski ⓘ https://orcid.org/0000-0002-0774-5983
Kate Bridges ⓘ https://orcid.org/0000-0003-3642-7068
Will Krause ⓘ http://orcid.org/0000-0001-7585-5749
Maria Fernanda Forni ⓘ https://orcid.org/0000-0002-3335-9023
Ryan R Driskell ⓘ http://orcid.org/0000-0001-7673-2564
Kathryn Miller-Jensen ⓘ http://orcid.org/0000-0002-7233-0100
Valerie Horsley ⓘ https://orcid.org/0000-0002-1254-5839

## Ethics

Diabetic and non-diabetic adults with chronic foot ulcers that were undergoing skin wound debridement were consented to donate discarded tissue for this study (IRB approval # 1609018360).
Mice were maintained through routine breeding in an Association for Assessment and Accreditation of Laboratory Animal Care (AALAC)-accredited animal facility at Yale University (Protocol # 11248). Animals were maintained on a standard chow diet ad libitum (Harlan Laboratories, 2018S) in 12-hour light/dark cycling. Up to five injured mice were housed per cage. All experimental procedures were approved and in accordance with the Institutional Animal Care and Use Committee.

## Decision letter and Author response

Decision letter https://doi.org/10.7554/eLife.86269.sa1
Author response https://doi.org/10.7554/eLife.86269.sa2

# Additional files

## Supplementary files

• Supplementary file 1. Demographics as reported in subjects' medical records.

• MDAR checklist

## Data availability

Sequencing data have been deposited in GEO under accession codes GSE223660 and GSE245703. The code to reproduce analyses of mouse scRNA-seq are available at DOI: 10.5281/zenodo.7562716.

The following datasets were generated:

| Author(s) | Year | Dataset title | Dataset URL | Database and Identifier |
|---|---|---|---|---|
| Horsley V, Phan Q, Schaefer-Sandoval T, King D | 2023 | Single Cell Analysis of Gene expression profile as single cell level of chronic diabetic foot ulcers and chronic non-diabetic foot ulcer Foot Wounds | https://www.ncbi.nlm.nih.gov/geo/query/acc.cgi?acc=GSE245703 | NCBI Gene Expression Omnibus, GSE245703 |
| Justynski O, Bridges K, Krause W, Forni MF, Phan Q, Sandoval-Schaefer T, Driskell R, Miller-Jensen K, Horsley V | 2023 | Apoptosis recognition receptors regulate skin tissue repair in mice | https://www.ncbi.nlm.nih.gov/geo/query/acc.cgi?acc=GSE223660 | NCBI Gene Expression Omnibus, GSE223660 |

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
