## [Editor Report]

The manuscript reports important new information on how dead cells are cleared from the wound site in order to promote effective repair of the damaged tissue. Interestingly, the authors find that the components of this clearance pathway are abnormally high in diabetics who have difficulty healing wounds and their results suggest that tampering them down may be a therapy to restore normal wound healing.

---

## [Decision Letter]

**Decision letter after peer review:**

Thank you for submitting your article "Apoptosis recognition receptors regulate skin tissue repair in mice" for consideration by *eLife*. Your article has been reviewed by 3 peer reviewers, one of whom is a member of our Board of Reviewing Editors, and the evaluation has been overseen by Satyajit Rath as the Senior Editor.

Essential revisions:

Overall, the reviewers are uniform in finding the manuscript a valuable contribution to the field and generally solid in methodology. However, there were a few consistent points that the technical rigor of the manuscript needs to be improved in order to better support the conclusions that are drawn from them. For instance, the suggested additional experiments are relatively straightforward and would substantially improve the data:

1. Quantify the apoptotic cells observed in the sections (Figure 1 I and J) and compare to control treatment on sections. It is not clear from the data presented whether the number of apoptotic cells increases or not in the time frame analyzed since the controls are lacking.

2. In a similar vein, it would be important to demonstrate in their system that there is actually an increase in apoptotic cells in the wounds. This data is absent and the inclusion of this would provide temporal insights into the initiation and speed of efferocytosis.

3. Conclusions based on CellChat analysis should be experimentally validated.

*Reviewer #1 (Recommendations for the authors):*

More recent datasets of efferocytosis related genes involved in diabetic wound healing such as SLC7 gene family members could be included in the analysis as it has direct bearing on the work.

The inhibition of Axl seems to have a dramatic effect scab formation, fibroblast representation in the wound bed, and vascularization but no significant effect on wound closure rates. This is a surprising observation and perhaps too much weight is given to ITGA6 staining and other assays of reepithelialization should be assessed. In addition, the potential therapeutic applications that the authors note for chronic wounds suggest that they should test the physiological parameters of the healed skin to determine how "normal" it truly is.

---

## [Author Response]

Essential revisions:Overall, the reviewers are uniform in finding the manuscript a valuable contribution to the field and generally solid in methodology. However, there were a few consistent points that the technical rigor of the manuscript needs to be improved in order to better support the conclusions that are drawn from them. For instance, the suggested additional experiments are relatively straightforward and would substantially improve the data:1. Quantify the apoptotic cells observed in the sections (Figure 1 I and J) and compare to control treatment on sections. It is not clear from the data presented whether the number of apoptotic cells increases or not in the time frame analyzed since the controls are lacking.

We thank the editors and reviewers for this important comment. We have repeated this staining experiment with the recommended antibody for cleaved (active) caspase 3. These data are now included in Figures 1H-J and Figure 1 – figure supplement 1F. While these were untreated samples, we were able to quantify the apoptotic cells visible by cleaved caspase 3 staining in 24h and 48h wounds. We found that, when quantified via IF staining, the apoptotic cells visible in the wound bed significantly increased between 24h and 48h after injury (Figure 1 – figure supplement 1F).

2. In a similar vein, it would be important to demonstrate in their system that there is actually an increase in apoptotic cells in the wounds. This data is absent and the inclusion of this would provide temporal insights into the initiation and speed of efferocytosis.

Thank you for this interesting and important suggestion. To analyze the initiation and speed of efferocytosis, we injected labeled apoptotic neutrophils into the wound bed at 1, 3, and 5 days after injury. 1 hour after this injection, we collected the wound beds for sectioning and staining and quantified the stained cells (either whole neutrophils or neutrophils undergoing efferocytosis) remaining in each wound bed. We found that significantly more neutrophils undergoing efferocytosis remained in 5D wound beds and that the rate of efferocytosis (calculated by dividing stained cells in efferocytosis by all stained cells) was constant over time, indicating that efferocytosis is active throughout the inflammatory and proliferative phases of wound healing. These data are now included in Figures 2H-M.

3. Conclusions based on CellChat analysis should be experimentally validated.

Thank you for this suggestion. We have compared the presence of Gas6 and Gas6^+^ macrophages in diabetic and nondiabetic human wounds via IF staining and found that, in accordance with the CellChat analysis, more macrophage Gas6 expression is observed in diabetic wounds. These data are included in Figures 3C-D.

Reviewer #1 (Recommendations for the authors):More recent datasets of efferocytosis related genes involved in diabetic wound healing such as SLC7 gene family members could be included in the analysis as it has direct bearing on the work.

Thank you for this helpful suggestion. We have included *Slc7a11* in the single-cell heatmaps in Figures 2A and Figure 3 – figure supplement 1A, as well as in the qPCR experiments in Figure 2 – Figure supplement 1B.

The inhibition of Axl seems to have a dramatic effect scab formation, fibroblast representation in the wound bed, and vascularization but no significant effect on wound closure rates. This is a surprising observation and perhaps too much weight is given to ITGA6 staining and other assays of reepithelialization should be assessed. In addition, the potential therapeutic applications that the authors note for chronic wounds suggest that they should test the physiological parameters of the healed skin to determine how "normal" it truly is.

Thank you for this suggestion. We have altered the text in lines 307-309 to specify the phenotype indicated by ITGA6 staining to give less weight to this result. We have also altered the sentence about therapeutic applications in lines 419-422 to indicate the need for further research.